# Patient Satisfaction with Aesthetic Outcomes Following OnabotulinumtoxinA Treatment for Chronic Migraine: A Cross-Sectional Study

**DOI:** 10.3390/toxins17060292

**Published:** 2025-06-08

**Authors:** Magdalena Boczarska-Jedynak, Marta Bott-Karoń, Karol Marschollek, Mariola Antolak, Maciej Świat, Marta Waliszewska-Prosół

**Affiliations:** 1Department of Neurology and Restorative Medicine, Health Institute dr Boczarska-Jedynak, 32-600 Oświęcim, Poland; 2Department of Neurology, Medicam Specialist Hospital, 72-300 Gryfice, Poland; 3Department of Neurology, Wroclaw Medical University, 50-556 Wroclaw, Poland; 4Neurology Department with Stroke Unit, Regional Specialist Hospital, 44-200 Rybnik, Poland; 5Department of Neurology, Faculty of Medicine, Academy of Silesia, 40-555 Katowice, Poland

**Keywords:** aesthetic medicine, onabotulinumtoxinA, chronic migraine, PREEMPT paradigm, wrinkles

## Abstract

OnabotulinumtoxinA (OnaBoNT-A) is approved for chronic migraine prevention and follows the PREEMPT protocol with injections in the glabellar and forehead regions. While aesthetic changes are considered a side effect, their effect on patient satisfaction has not been thoroughly assessed. This study evaluated patient satisfaction with facial aesthetic outcomes after repeated OnaBoNT-A treatment for chronic migraine. Conducted at specialist headache centers, it included adult patients with chronic migraine who had received at least three OnaBoNT-A cycles. Participants completed a structured questionnaire on demographics, migraine history, facial wrinkles and age perception, appearance satisfaction, psychological impact, treatment satisfaction, and adverse aesthetic events. A total of 124 patients (92.7% female; median age 42.5 years) participated. OnaBoNT-A reduced wrinkle severity (*p* < 0.0001). Most patients (74.2%) reported aesthetic improvement post-treatment. The majority of patients (76.7%) declared that treatment met or exceeded expectations. 32% reported looking younger post-treatment, with a median perceived age difference of 5 years. Adverse event frequency was similar to pivotal trial outcomes, mostly mild, with no treatment discontinuations. OnaBoNT-A for chronic migraine, following the PREEMPT protocol, provides significant therapeutic benefits and high patient satisfaction regarding aesthetic outcomes. Although aesthetic side effects were generally mild, they were not uncommon.

## 1. Introduction

Botulinum toxin type A (BoNT-A) is an effective therapeutic agent used across different dermatologic and neurologic indications [1]. It exerts its primary effect by temporarily paralyzing muscles through inhibition of acetylcholine release. In addition to its neuromuscular effects, BoNT-A also exhibits antinociceptive properties by inhibiting the release of pain mediators such as calcitonin gene-related peptide, substance P, and vasoactive intestinal peptide from sensory nerves [2]. Aesthetic indications were first approved by the U.S. Food and Drug Administration (FDA) for glabellar lines in 2002, followed by lateral canthal lines in 2013, and forehead lines in 2017 [3].

The therapeutic effects of BoNT-A in headache management were first observed in the nineties, not by neurologists but by aesthetic physicians who noted headache relief in patients undergoing treatment for facial wrinkles [4]. In 2010, onabotulinumtoxinA (onaBoNT-A; BOTOX^®^, Allergan, Irvine, CA, USA) received U.S. Food and Drug Administration (FDA) approval for the treatment of CM, and it remains the only BoNT-A approved for the prophylactic management of this disabling condition [2]. The only approved injection protocol for the treatment of CM with onaBoNT-A is the PREEMPT (Phase III REsearch Evaluating Migraine Prophylaxis Therapy) paradigm. This protocol involves the administration of 155–195 units (U) of onaBoNT-A across 31–39 injection sites, with 5 U delivered per site, targeting seven specific muscle groups of the face, head, and neck, including the glabellar and forehead regions [5,6,7]. In the upper face, this procedure resides at the intersection of neurology and aesthetic medicine [8]. Although aesthetic changes are considered side effects of migraine therapy [9], patient satisfaction with appearance or dissatisfaction may affect treatment adherence. Proper injection technique, tailored to individual anatomical variations, is crucial for minimizing aesthetic complications such as eyelid ptosis, brow ptosis, or the Mephisto effect. Recently, guidelines have been published to optimize the aesthetic outcomes of the PREEMPT protocol, with a particular focus on injection techniques in the upper face [8,10].

Given the frequency of forehead asymmetry, eyelid, and brow ptosis among patients receiving onaBoNT-A for CM [9], it is reasonable to assume that some individuals may be dissatisfied with their post-treatment appearance but continue therapy due to its efficacy in reducing migraine symptoms. Conversely, others may express high satisfaction with their appearance following treatment, particularly due to improved glabellar lines and forehead wrinkles. Despite these observations, the aesthetic impact of onaBoNT-A treatment in CM patients has not been systematically investigated. This study aimed to assess the level of patient satisfaction with aesthetic outcomes following a minimum of three treatment cycles of onaBoNT-A for CM. The study addressed how patients perceive changes in their facial appearance, including subjective assessments of wrinkle severity and overall aesthetic improvement, after onaBoNT-A treatment based on the PREEMPT paradigm.

## 2. Results

### 2.1. Patient Population

A total of 124 patients were enrolled, with women comprising the majority of respondents (Table 1). Most patients were of Caucasian ethnicity with IV or V Fitzpatrick scale phototypes. The participation rate among patients fulfilling the inclusion criteria was 82.0%. All patients had a minimum of a few years history of CM (Table 1). The majority of patients (*n* = 90, 72.6%) received three treatment cycles according to the PREEMPT paradigm (Table 1). Overall, in 105 patients (84.7%), onaBoNT-A injections were administered by the same physician. The treatment was fully financed from public sources in 82 patients (66.1%); otherwise, it was financed out of pocket or performed during workshops. Over 80% of patients reported having at least one comorbidity to CM, and the most frequent conditions and diseases (occurring in ≥10% of patients) are listed in Table 1. Sixteen patients (12.9%) reported previous treatment with BoNT-A for aesthetic purposes.

The median score for the effect of treatment on CM was 5 (range, 0–10), with 80/123 patients (65.0%) reporting a score ≥3 and 68/123 patients (55.3%) reporting a score ≥5.

### 2.2. Primary Outcome

CM treatment with onaBoNT-A had a positive outcome for patient-reported aesthetic improvement of glabellar frown line (FL) and horizontal forehead wrinkles (FW), as evaluated at rest and maximal contraction. The median FW severity at rest before treatment was 1.5 (range, 0–3) and decreased to 0 (range, 0–3) after treatment (*p* < 0.0001) (Figure 1A), with an increase reported only in one case. At maximal frown, the median FL severity was 2 (range, 0–3) before and 1 (range, 0–3) after treatment (*p* < 0.0001) (Figure 1B). After treatment, 9.2% (11/120) and 13.9% (17/122) of patients reported moderate or severe FL at rest or maximum frown, respectively. A similar pattern of changes after treatment was observed in the FW severity assessment; the median FW severity changed from 1 to 0 (range, 0–3) at rest and from 2 to 1 (range, 0–3) at maximal contraction (*p* < 0.0001) (Figure 1C,D). After treatment, 11.6% (14/121) and 17.6% (21/119) of patients reported moderate or severe FW at rest or maximum contraction, respectively. There were no statistically significant differences in FL and FW between patients aged <40 and ≥40 years old (*p* > 0.1).

The satisfaction rate regarding FL and FW improvement after treatment for CM was high, with the majority of patients reporting slight to very high improvement or no change. Only 2% of patients reported a slight worsening of FL and 4% of FW. None of the patients reported a significant worsening (Figure 2).

### 2.3. Secondary Outcomes

Two-thirds of patients reported that they look their current age (82/122, 67.2%). Thirty-nine patients (32.0%) considered themselves to look younger than their current age, indicating a median age difference of 5 years (range, 2–15 years) younger than their actual age. Only one patient reported that she appeared to be 5 years older than her current age.

Patients were asked to evaluate their satisfaction with treatment using the Facial Line Satisfaction Questionnaire (FLSQ), which was developed to assess overall satisfaction, treatment effectiveness, discomfort or side effects, convenience of treatment, ease of treatment, flexibility, and time to onset of treatment for facial lines. Figure 3 presents the assessment results, providing median scores for each item along with 95% CI. Patient satisfaction with the aesthetic effect of the treatment was high, with a median score of 1 (rather satisfied), across all FLSQ satisfaction domain items (Figure 3A). The aesthetic effect of the treatment was rated as meeting expectations in 68 out of 120 patients (54.7%) and as exceeding expectations in 24 out of 120 patients (20.0%). The remaining patients (28/120, 23.3%) declared that the treatment did not meet their expectations. In the FLSQ Impact Domain, higher scores indicate a greater negative impact on the patient. Most patients reported that facial lines have a slight or no effect on them (Figure 3B). Only 11.4% of patients (14/123) reported a negative impact in any of the five domain items (scores 4 or 5).

The psychological impact of upper facial lines after treatment with onaBoNT-A for CM was assessed using the Facial Line Outcomes (FLO-11) questionnaire. Figure 4 summarizes median scores with 95% CI reported for each item scored from 0 to 10, where a higher score indicates a larger negative impact, except feeling good about facial appearance, which was rated reversely. The median total score was 26 (range, 0–90; 95% CI, 18.2–31.0). The percentage of patients declaring a score of ≥5 for any of the items, except the last one, which was rated reversely, was 48.7% (59/121).

### 2.4. Association Between the Effect of Treatment on Chronic Migraine and Satisfaction with Facial Appearance

The effect of treatment of CM was not associated with the psychological impact of upper facial lines after treatment with onaBoNT-A, measured by the FLO-11, and the impact domain of the FLSQ. The r coefficient for the correlation between the effect on migraine and FLO-11 was 0.0969 (95% CI, −0.0846–0.2722), *p* = 0.2944. The r coefficient for the correlation between the effect on migraine and the impact domain of FLSQ was 0.1238 (95% CI, −0.0551–0.2951), *p* = 0.1238.

### 2.5. Safety of Chronic Migraine Treatment with OnabotulinumtoxinA

Overall, 34 patients (27.4%) reported the injection procedure, according to the PREEPMT paradigm, to be uncomfortable, including nine patients describing it as very uncomfortable. Table 2 summarizes all adverse events (AEs) that occurred in the upper face at any time during multiple rounds of treatment of CM with onaBoNT-A. A minimum of one AE was reported by 52 patients (41.9%). In addition to aesthetic AEs, patients reported pain after injections (*n* = 2) and migraine intensification (*n* = 1). Among 69 AEs reported, treating physicians were informed about only 19 AEs (27.5%). The most common reasons for non-reporting were a low intensity of AEs (*n* = 14) and lack of awareness of AE association with OnaBoNT-A action (*n* = 12). None of the AEs led to treatment discontinuation, and all patients who participated in the study declared their willingness to continue the treatment.

## 3. Discussion

This is the first study to assess patient-perceived aesthetic benefits of CM treatment with onaBoNT-A. In addition to reducing migraine frequency, the onaBoNT-A injections, according to the PREEMPT protocol, can yield cosmetic benefits in the glabellar and forehead areas, translating into high patient satisfaction with appearance and well-being.

The migraine-relieving potential of BoNT-A was first observed by aesthetic medicine physicians treating glabellar frown lines. Later, through rigorous trials designed to maximize efficacy in migraine prevention, the PREEMPT protocol was developed. The paradigm initially did not prioritize aesthetic considerations; however, in the upper face, the treatment procedure inherently intersects the fields of neurology and aesthetic medicine. Later experiences have allowed for the tailoring of the injection technique to individual patient anatomy, which may help prevent undesired aesthetic outcomes in the glabellar and forehead regions.

In the studied cohort, patients reported an improvement in the severity of FL and FW. These observations are consistent with prior research in purely aesthetic patient populations, where a BoNT-A treatment of glabellar and forehead lines resulted in both objectively and subjectively smoother skin and high patient satisfaction [11,12,13,14]. The patient population in our study was more heterogeneous based on declared wrinkle severity, i.e., included patients with no or mild wrinkles before treatment. Studies recruited patients only with moderate-to-severe wrinkles at maximal contraction [11,12,13,14]. Response rates, the proportion of patients with none or mild wrinkles one month after treatment, ranged from 60 to 80% [11,12,13,14]. In our study, only a minority of patients who received a BoNT-A treatment for CM would qualify for treatment in an aesthetic indication according to eligibility criteria from phase 3 trials. The majority reported improvement in the severity of upper facial lines and noted overall improvement in their aesthetic appearance after treatment for CM, especially in FL.

The present findings confirm that, in addition to reducing migraine frequency, the standard PREEMPT protocol can yield appreciable aesthetic benefits in the glabellar and forehead areas, which translate into high patient satisfaction with appearance. In a randomized, double-blinded trial, 80% of patients with moderate to severe FL were satisfied with their facial appearance after a BoNT-A treatment as evaluated using the FLSQ. This benefit was sustained for up to 4 months [15]. Since injections of onaBoNT-A inherently overlap the areas of neurology and aesthetic medicine, we suspected that a high level of satisfaction and psychological well-being could be observed as a result of CM treatment. Prominent frown or forehead lines may make a patient feel unattractive or older than their current age, and the psychological burden of upper facial rhytids has been documented in qualitative studies [16]. Our results suggest that patients with CM may experience similar psychosocial benefits from the aesthetic effects of treatment; however, due to the cross-sectional design and reporting of FLO-11 and FLSQ as changes upon treatment, it is challenging to compare the outcomes achieved in aesthetic medicine studies [15,17,18,19]. Nevertheless, both these scores reached levels suggesting a lower to moderate perceived negative impact of facial lines, indicating that patients are emotionally and socially comfortable with their appearance. Only a small subset of patients experienced psychological concern from the appearance of facial lines, underscoring the need for individualized injection strategies. The effect of the treatment on CM was not correlated with satisfaction with facial appearance.

In our study, the assessments were performed after three cycles of treatment with onaBoNT-A. The IHS guidelines suggest evaluating response after a minimum of three months for monthly injectable drugs and six months for quarterly injections [20]. In the PREEMPT trial, approximately one-third of responders achieve meaningful benefits only in the second and third cycles [21]. Therefore, we assumed that assessment after three cycles reflects improvements in headache frequency, intensity, and quality of life. The former improves progressively over multiple treatment cycles. The assessment after minimum tree treatment cycles aimed to ensure the capture of broader benefits.

Balanced against these benefits, aesthetic AEs that can occur with BoNT-A injections in the upper face are well known, i.e., an elevated risk of ptosis and muscle weakness [22]. Our study documents a spectrum of events that can occur at any time during treatment, and their incidence sheds light on the challenges of applying a fixed migraine protocol to diverse individual anatomies. The most frequently reported event was forehead movement discrepancy, which refers to reduced forehead expression, i.e., facial masking. However, facial masking and eyebrow asymmetry are usually mild; more conspicuous AEs include eyelid and brow ptosis, which were reported at rates similar to those observed in PREEMPT trials [23]. Brow ptosis likely results from the inadvertent spread of toxin into the frontalis muscle’s inferior fibers, thereby weakening eyebrow elevation [8,22]. Sometimes, forehead injection or treating the glabellar complex without adequately addressing lateral forehead activity can create the opposite imbalance—a pronounced lateral eyebrow lift, commonly known as the Mephisto sign. This was reported in 2.4% of patients, which aligns with anecdotal reports among injectors; however, it is rarely reported in trials. Along with eyelid/brow ptosis, it represents a spectrum of aesthetic AEs that, while not medically dangerous, can be distressing to patients. It is noteworthy that in our cohort, a large fraction of patients experienced some aesthetic AE during repeated treatments. Most of these events were mild and transient, and importantly, many patients did not spontaneously report them to their physicians, often because they did not realize the issue was treatment-related or felt it was not severe enough to mention. None of the AEs caused treatment discontinuation, and all patients declared their willingness to continue the treatment for CM. Nonetheless, the occurrence of even minor aesthetic complications is clinically relevant and may affect treatment adherence.

The glabellar and forehead muscles are central not only to migraine pathophysiology [23] but also to facial expression and appearance. This dual role presents unique challenges when the patient is treated in both neurologic and aesthetic indications. Recent literature has begun to address these technical and anatomical considerations, proposing a refined injection approach for the upper face in chronic migraine patients, aiming to optimize aesthetic outcomes without sacrificing headache control. Considering the significant effect of CM treatment with onaBoNT-A on patients’ appearance and well-being, establishing best practices that balance neurology and aesthetics is essential [8].

### Limitations of the Study

The study has multiple limitations. It used a cross-sectional, single-timepoint survey approach, which limits the ability to assess causality or changes in aesthetic satisfaction over time. It does not allow for a comparison of patient perceptions before and after each treatment cycle or an evaluation of the long-term aesthetic impact of onaBoNT-A. Patients were asked to recall the severity of their facial lines before treatment. This introduces recall bias, as participants may inaccurately remember their past appearance, especially if they have been receiving treatment for an extended period. The lack of a control group limits the ability to determine the extent to which observed aesthetic changes and satisfaction are attributable to the treatment. Although most patients were treated according to the PREEMPT paradigm, the involvement of different injectors and variation in the number of treatment cycles may have influenced the results. The evaluation relied on patient-reported outcomes without any objective clinical or photographic assessment of wrinkle severity or facial appearance. The study population was predominantly female and recruited from selected specialist centers. Thus, the findings may not be generalizable to male patients.

## 4. Conclusions

This study highlights the significance of patient-perceived aesthetic outcomes in the long-term management of CM with BoNT-A therapy. In adult patients with CM, onaBoNT-A injections according to the PREEMPT paradigm not only significantly reduced glabellar and forehead wrinkle severity but also yielded high aesthetic satisfaction. Although a subset of patients experiences adverse cosmetic effects, these outcomes are typically predictable and can be mitigated with precise injection techniques. Integrating principles of aesthetic medicine into migraine treatment protocols may optimize therapeutic outcomes, offering both symptom relief and the psychological benefits associated with a relaxed, natural appearance dual, neurologic, and aesthetic benefits may enhance long-term adherence.

## 5. Materials and Methods

This cross-sectional, non-interventional study was conducted from December 2024 to March 2025 at four clinical centers specializing in the treatment of migraine in Poland. The study was approved by the Ethical Committee of the Medical Chamber in Bielsko-Biała (reference number 2024/11/21/2). All participants gave written informed consent to participate.

### 5.1. Patients

The study included adult patients diagnosed with chronic CM in accordance with the criteria of the International Classification of Headache Disorders-3 (ICHD-3) [24]. All patients had to receive a minimum of three courses of treatment with onaBoNT-A according to the PREEMPT paradigm as part of CM management. The treatment included a series of intramuscular and subcutaneous injections, each of 5 U of onaBoNT-A, into 39 injection sites in the face, head, and neck. All patients received a total dose of 195 U of onaBoNT-A, including the follow-the-pain approach. Cycles of onaBoNT-A were administered every 12 weeks. All injection sites and administered doses were stable and symmetrical. None of the patients received additional doses of onaBoNT-A for aesthetic purposes during the treatment period. Exclusion criteria included the presence of other primary or secondary headache disorders, comorbid chronic pain syndromes, or severe psychiatric or systemic illnesses. All constitutive patients attending regular visits to participating sites and meeting the eligibility criteria were invited to participate. The participation was unconditional and without incentive. Patients consenting to participate completed a structured questionnaire consisting of seven main sections: demographic information, medical history, details of onaBoNT-A treatment, self-perception of wrinkles at the time of assessment and retrospectively before treatment, subjective age perception, patient-reported outcome measures assessing treatment effectiveness and satisfaction with facial appearance, and history of treatment-related AEs.

### 5.2. Outcomes

The primary endpoint was the evaluation of onaBoNT-A, used according to the PREEMPT paradigm, on facial expression lines, including the glabellar FL and horizontal FW. The assessment of FL and FW before and after onaBoNT-A treatment was established using Likert scales: the severity of wrinkles—using to a 4-point categoric scale: no wrinkles (0), mild wrinkles (1), moderate wrinkles (2), and severe wrinkles (3) based on the Facial Wrinkle Grading System [25] and the FACE-Q conceptual framework [26]. The satisfaction rate was evaluated based on the Global Aesthetic Improvement Scale (GAIS), a 7-point scale that measures the post-treatment improvement rate (very much improved, much improved, slightly improved, no change, slightly worsened, much worsened, and very much worsened).

Secondary outcomes included patient-reported subjective age change compared to the period before treatment with onaBoNT-A, psychological impact, satisfaction with facial appearance, and well-being following treatment. The age change perception was evaluated using three statements: “I look my current age”, “I look younger”, and “I look older”. The psychological impact of upper facial lines was evaluated using the FLO-11 questionnaire [16,27], and patient satisfaction with the aesthetic effect of CM treatment was assessed using the FLSQ [15,28,29]. Both tools were previously used in clinical trials assessing the effects of onaBoNT-A and were developed in accordance with FDA PRO guidance [30,31].

The effect of treatment on CM was evaluated by patients on a scale of 0 to 10, where 0 indicated no effect, and 10 indicated full relief. To assess associations between the effect on CM and satisfaction with facial appearance, Pearson correlations were computed between each declared effect on migraine and the total scores of the FLO11 or the impact domain of the FLSQ. Finally, patients answered questions about possible aesthetic AEs of the therapy, such as a feeling of inability to lift the forehead, lowering of both eyebrows (drooping forehead), asymmetry of eyebrow position—brow ptosis, medial brow ptosis, and lifting of the lateral parts of the eyebrows (Mephisto sign), real eyelid ptosis, or others. For each AE reported, patients were asked whether they had informed their treating physician about it. If not, the reasons for non-disclosure were explored.

### 5.3. Data Analysis

There was no formal calculation of the sample size because differences in injection technique and doses of onaBoNT-A used in the PREEMPT protocol and aesthetic treatment of the glabellar and forehead regions did not allow extrapolation of the effect on the severity of wrinkles from aesthetic to neurologic treatment. Similarly, a review of the literature on the safety of onaBoNT-A treatment for CM did not reveal any information about worsening wrinkle severity. It was impossible to assume any direction of the effect of onaBoNT-A treatment for CM on FL and FW severity. This topic was not studied previously.

The Shapiro–Wilk method and visual inspection were used to assess the normality of data. No imputation for missing values was performed. Continuous variables were described through medians with range or 95% confidence intervals, whereas categorical variables were presented as frequencies with percentages. Changes in the severity of wrinkles were compared with the Wilcoxon test, and differences between patients aged <40 and ≥40 years old were analyzed using the Mann–Whitney test. A *p*-value < 0.05 was regarded as statistically significant for all tests. Data analysis was performed using MedCalc ver. 23.1.3 (MedCalc software Ltd., Ostend, Belgium).

## Figures and Tables

**Figure 1 toxins-17-00292-f001:**
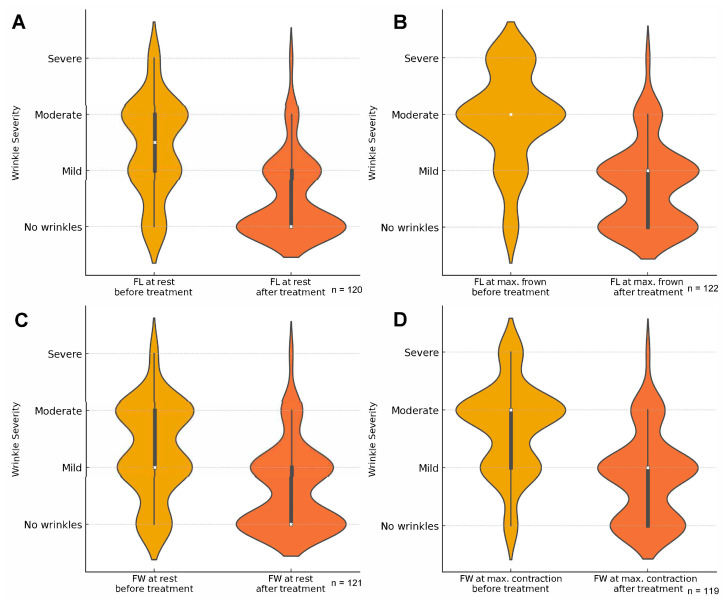
Violin plots of changes in wrinkle severity declared by patients in the glabellar frown line at rest (**A**), at maximal frown (**B**), in horizontal forehead wrinkles at rest (**C**), and maximal contraction (**D**).

**Figure 2 toxins-17-00292-f002:**
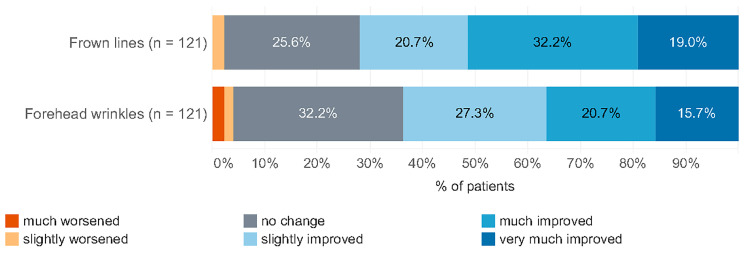
Global Aesthetic Improvement Scale assessment.

**Figure 3 toxins-17-00292-f003:**
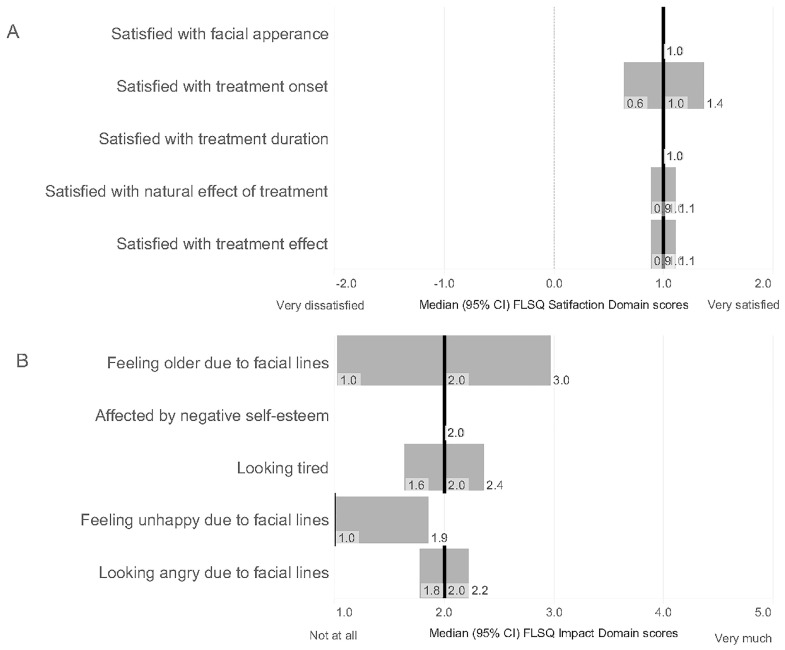
Facial Line Satisfaction Questionnaire assessment, including satisfaction domain items (**A**) and impact domain items (**B**). Results were presented as medians (black lines) with 95% CI.

**Figure 4 toxins-17-00292-f004:**
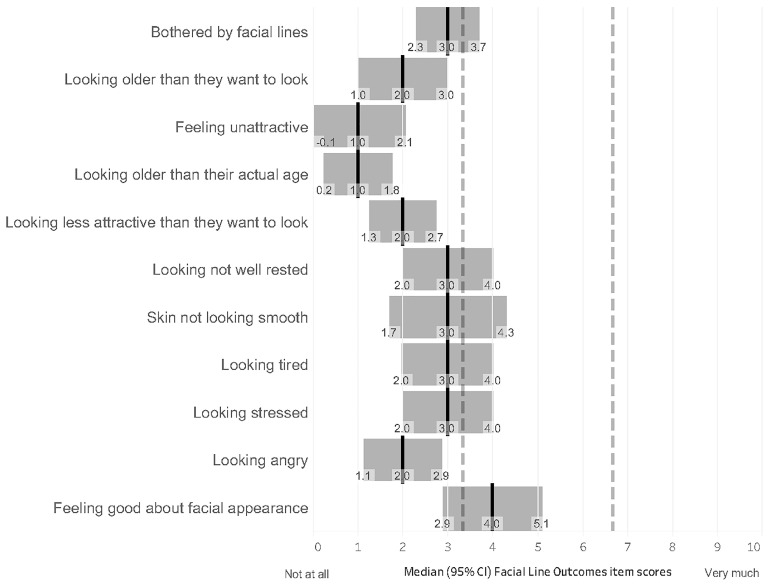
Facial Line Outcomes assessment. Results were presented as medians with 95% CI. Black solid lines indicate medians, and dotted lines were used to divide the scale into three equal parts.

**Table 1 toxins-17-00292-t001:** Patient characteristics.

Characteristic	*N* = 124
Sex, *n* (%)	
Female	115 (92.7)
Male	9 (7.3)
Median age (range), years	42.5 (21.0–72.0)
Median body mass index (range)	23.6 (16.8–35.4)
Overweight (BMI > 30), *n* (%)	34 (27.4)
Obesity (BMI > 30), *n* (%)	9 (7.3)
Median age at first migraine episode (range), years	16.5 (6.0–45.0)
Median disease duration (range), years	23.0 (6.0–62.0)
Median number of onaBoNT-A treatment cycles (range)	3 (3–10)
Any comorbidity, *n* (%)	103 (83.1)
Tension-type headache	50 (40.3)
Anxiety	20 (16.1)
Bruxism	20 (16.1)
Hypothyroidism	20 (16.1)
Depression	19 (15.3)
Insomnia	14 (11.3)

onaBoNT-A, onabotulinumtoxinA; BMI, body mass index.

**Table 2 toxins-17-00292-t002:** Aesthetic adverse events of onabotulinumtoxinA treatment for chronic migraine.

Adverse Events, *n* (%)	*N* = 124
Forehead movement discrepancies	36 (29.0)
Eyebrow asymmetry	9 (7.3)
Eyelid ptosis	7 (5.6)
Brow ptosis	5 (4.0)
Mefisto sign	3 (2.4)
Crow’s feet discrepancies	2 (1.6)
Forehead wrinkles	1 (0.8)

## Data Availability

Academic study groups can request de-identified patient data, indicating the aim of the request.

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
