# Peer review of "Patient Satisfaction with Aesthetic Outcomes Following OnabotulinumtoxinA Treatment for Chronic Migraine: A Cross-Sectional Study"

_toxins, 2025, doi:10.3390/toxins17060292_

Round 1
Reviewer 1 Report
Comments and Suggestions for Authors
The authors reported the aesthetic benefits of botulinum injection for chronic migraine
The aesthetic improvement was not unexpected.
The adverse effect of the injections included some asymmetry of facial wrinkles. It would be interesting to know if the injection points and dosage over the face were symmetrical in the first place.
Author Response
Comment 1: The adverse effect of the injections included some asymmetry of facial wrinkles. It would be interesting to know if the injection points and dosage over the face were symmetrical in the first place.
Response 1: Correct, the asymmetry was reported by some patients. It was typically mild. We confirm that the injection points and doses administered were symmetrical and stable during the whole treatment period, what was included in the Methods section after revision. We also indicated in the limitations part of the Discussion that injections were performed by different injectors, and variation in the number of treatment cycles may have influenced the results.
Reviewer 2 Report
Comments and Suggestions for Authors
In this interesting study, authors evaluated patient satisfaction with facial aesthetic outcomes after repeated OnaBoNT-A treatment for chronic migraine.
Although study is original because it evaluates a secondary aesthetic aspect of the treatment of chronic migraine, because it is conducted with a precise statistical evaluation and provides conclusions in line with the results, I believe that the manuscript needs some important revisions before published.
My comments:
1)in materials and methods section it is necessary to specify the average total and specific dosage on the muscles of the facial region used (in the manuscript minimum/maximum of international protocol dosages are cited)
2)create a subparagraph "limitations of the study" in the discussion section
3)improve the bibliography as only 8/21 are under 5 years old
4) minor improve English
Minor improve English
Author Response
Comment 1: In the materials and methods section it is necessary to specify the average total and specific dosage on the muscles of the facial region used (in the manuscript minimum/maximum of international protocol dosages are cited)
Response 1: Thank you. We added a description in the Methods section. In our study, all patients received the follow-the-pain series of the PREEMPT protocol intramuscular and subcutaneous injections. Thus, the total dose was 195 U of onaBoNT-A/ patient in each treatment cycle.
Comment 2: create a subparagraph "limitations of the study" in the discussion section.
Response 2: Thank you. This paragraph was included in the Discussion section (point 3.1)
Comment 3: improve the bibliography as only 8/21 are under 5 years old.
Response 3: Thank you for the comment. The references in the study were selected based on their relevance, not the year of publication. The years of publication of selected articles come from the fact that both aesthetic and neurologic indications of onaBoNT-A are relatively old, and the pivotal studies, including those using outcomes studied by us, were published a while ago. There are no articles addressing the topic of our study up to date. That is the reason we decided to conduct this study.
Comment 4: minor improve English
Response 4: We revised the manuscript. All changes were introduced based on comments, and language improvements are tracked.
Reviewer 3 Report
Comments and Suggestions for Authors
1. Why is skin color not included as a variable in the study design?
2.In the discussion section, there is no analysis of the correlation between variables such as age and sex in relation to the study findings on migraine.
3.The study’s recommendations should be clearly reported.
4.The basis for determining the sample size should be stated.
5.The inclusion and exclusion criteria should be clearly mentioned in the Methods section.
6.Grammar and punctuation throughout the manuscript should be revised.
7.The references need to be updated.
Author Response
Comment 1: Why is skin color not included as a variable in the study design?
Response 2: Thank you. All patients who participated in the study were white (Caucasian ethnicity) and Fitzpatrick phototype IV or V. We included this information in the manuscript's Results section. However, it is not possible to include is as a variable.
Comment 2: In the discussion section, there is no analysis of the correlation between variables such as age and sex in relation to the study findings on migraine.
Response 2: We agree with the comment. We divided the population according to age (<40 and ≥40 years old). We compared the primary outcome according to age and found no differences (P > 0.05 for the Mann-Whitney test). We included this information in the updated results and methods sections. We confirm that there are differences in the FLO-11 scale, too; however, they are expected, i.e., older patients had a median of 1 or 2 points higher impact than the younger for almost every item of the scale (P < 0.05). These medians are still within the 95% CI for the entire population. Thus, we did not include these results in the results section. There are no differences in the FLSQ scale. Regarding the sex, males consist only of a minority (<10%), and the analysis can not be performed.
Comment 3: The study’s recommendations should be clearly reported.
Response 3: We revised the conclusion section of the manuscript for better clarity.
Comment 4: The basis for determining the sample size should be stated.
Response 4: We performed an attempt to establish the sample size based on the literature data. However, we failed. Injections of the PREEMPT protocol and for aesthetic purposes are very different, which makes it challenging to extrapolate outcomes from aesthetic studies, with outcomes of our study, to patients with chronic migraine. This was described in the Methods section (point 2.3.). We decided to enroll all the constitutive patients treated in our sites during following 3 months.
Comment 5: The inclusion and exclusion criteria should be clearly mentioned in the Methods section.
Response 6: Methods section indicated the following inclusion criteria: diagnosis of chronic migraine and minimum three cycles of treatment with ona-BoNTA according to the PREEMPT protocol, and the following exclusion criteria: the presence of other primary or secondary headache disorders, comorbid chronic pain syndromes, or severe psychiatric or systemic illnesses.
Comment 6: Grammar and punctuation throughout the manuscript should be revised.
Response 6: We revised the manuscript. All changes were introduced based on comments, and language improvements are tracked.
Comment 7: The references need to be updated.
Response 7: Thank you for the comment. The references in the study were selected based on their relevance, not the year of publication. The years of publication of selected articles come from the fact that both aesthetic and neurologic indications of onaBoNT-A are relatively old, and the pivotal studies, including those using outcomes studied by us, were published a while ago. There are no articles directly addressing the topic of our study up to date. That is the reason we decided to conduct this study.
Round 2
Reviewer 2 Report
Comments and Suggestions for Authors
Authors improved manuscript according to my opinion.
No other comments
Author Response
Thank you.
Reviewer 3 Report
Comments and Suggestions for Authors
Thanks
Author Response
Thank you